# Simulation and Experimental Investigation of Balloon Folding and Inserting Performance for Angioplasty: A Comparison of Two Materials, Polyamide-12 and Pebax

**DOI:** 10.3390/jfb14060312

**Published:** 2023-06-05

**Authors:** Tao Li, Zhuo Zhang, Wenyuan Wang, Aijia Mao, Yu Chen, Yan Xiong, Fei Gao

**Affiliations:** 1College of Mechanical Engineering, Sichuan University, Chengdu 610065, China; 2Chengdu Neurotrans Medical Technology Co., Ltd., Chengdu 610065, China; 3Department of Applied Mechanics, Sichuan University, Chengdu 610065, China

**Keywords:** angioplasty, balloon dilatation catheter, finite element analysis, bench test, insertion force, balloon pleating simulation

## Abstract

Background: A balloon dilatation catheter is a vital tool in percutaneous transluminal angioplasty. Various factors, including the material used, influence the ability of different types of balloons to navigate through lesions during delivery. Objective: Thus far, numerical simulation studies comparing the impacts of different materials on the trackability of balloon catheters has been limited. This project seeks to unveil the underlying patterns more effectively by utilizing a highly realistic balloon-folding simulation method to compare the trackability of balloons made from different materials. Methods: Two materials, nylon-12 and Pebax, were examined for their insertion forces via a bench test and a numerical simulation. The simulation built a model identical to the bench test’s groove and simulated the balloon’s folding process prior to insertion to better replicate the experimental conditions. Results: In the bench test, nylon-12 demonstrated the highest insertion force, peaking at 0.866 N, significantly outstripping the 0.156 N force exhibited by the Pebax balloon. In the simulation, nylon-12 experienced a higher level of stress after folding, while Pebax had demonstrated a higher effective strain and surface energy density. In terms of insertion force, nylon-12 was higher than Pebax in specific areas. Conclusion: nylon-12 exerts greater pressure on the vessel wall in curved pathways when compared to Pebax. The simulated insertion forces of nylon-12 align with the experimental results. However, when using the same friction coefficient, the difference in insertion forces between the two materials is minimal. The numerical simulation method used in this study can be used for relevant research. This method can assess the performance of balloons made from diverse materials navigating curved paths and can yield more precise and detailed data feedback compared to benchtop experiments.

## 1. Introduction

Peripheral artery disease (PAD) is a long-term arterial disease caused by atherosclerosis in the peripheral vessels, leading to complications in limbs such as pain, ulcers, gangrene, and reduced function [1]. Percutaneous transluminal coronary angioplasty (PTCA) is an effective approach for treating symptomatic atherosclerotic peripheral artery disease [2]. This disease affects a wide patient population worldwide, with more than 200 million individuals afflicted [3]. The balloon-tipped catheter is a device delivered to the site of a lesion, and it functions to increase the lumen’s diameter [4]. The roles played by different types of balloons in treatment vary greatly [5,6,7]. For instance, drug-coated balloons can directly deliver antiproliferative drugs to the local lesion in the blood vessel without the need for implanting a stent, while cutting balloons have metal blades attached to their surface that can cut through plaque during expansion. Additionally, different balloon structures, such as dual-wire and scoring balloons, cater to specific functional requirements. In terms of compliance, balloons can be categorized into compliant, semi-compliant, and non-compliant balloons. Compliant balloons adapt better to the morphology of the blood vessels, whereas non-compliant balloons have a superior ability to compress hard lesions. PTCA balloons are primarily composed of thermoplastic polymers, including polyethylene terephthalate (PET), nylon, polyethylene with additives, polyvinyl chloride (PVC), and polyurethane used along with nylon [8]. The materials used for PTCA balloons significantly affect their characteristics [9]. Currently, Polyamide 12 and Pebax are widely used in the market to manufacture PTCA balloons [10]. Trackability, which refers to the force needed to navigate a balloon catheter through a tortuous path to the target lesion [11], is a crucial indicator for evaluating the performance of PTCA balloons [12,13]. A lower value is preferred [14]. The trackability of a real balloon is tested in the folded state, and good folding properties, which determine the ease of folding and unfolding, also play a crucial role in maintaining the performance of the balloon after repeated inflations [12]. Therefore, the quality of the folded state directly affects the trackability of the balloon.

Finite element analysis (FEA) has become a powerful tool for the optimization process of coronary stents and balloon catheters. There have been many studies that have conducted experiments and simulations on balloons from various perspectives. Geith et al. [15] proposed a simple, microstructurally motivated constitutive model aimed at mimicking the pronounced anisotropic material response observed in the performed experiments. Helou et al. [16] presented a modeling method for simulating percutaneous transluminal angioplasty (PTA) endovascular treatment and evaluated the effects of balloon design, plaque composition, and balloon sizing on acute post-procedural outcomes after PTA. Dong et al. [17] investigated the efficacy of post-dilation balloon diameter and inflation pressure in improving the stent expansion in a calcified lesion. Rahinj et al. [18] analyzed a non-uniform balloon stent expansion pattern comprised of variations in the stent axial position on the balloon, balloon length, balloon folding pattern, and balloon wall thickness. De Beule et al. [19] proposed a trifolded balloon methodology, which was confirmed through experiments and agreed with manufacturers’ data. Wiesent et al. [20] performed a stent life-cycle simulation including balloon folding, stent crimping, and the free expansion of the balloon–stent system. Bukala et al. [21] deployed the “kissing balloon” stenting technique applied to patients with bifurcation stenosis. Hamed et al. [22] verified through experiments that the cross-sectional scanning morphology of a balloon under different pressures and the diameter curve under applied pressure can be consistent with a 2D simulation, demonstrating the consistency between simulation and experimental results. Gajewski et al. [23] demonstrated that material stiffness has a significant impact on the degree of occlusion of the balloon in the artery. The clear advantage of a contrasting engineering simulation is that it provides a comprehension evaluation of a design, offering insightful feedback and thereby minimizing the need for complex and expensive experiments, which are often difficult to carry out. 

However, to our knowledge, there have been no studies comparing simulated folding and tracking performances between two different materials. Although Sirivella et al. [11] successfully simulated the pushability and trackability of polyamide PTCA balloons in the folded state using finite element analysis and compared the results with experiments, they did not consider the pre-stress of the folded balloon. Instead, they directly established the initial model of the balloon in the folded configuration, which compromises some realism. On the other hand, Geith et al. [24] simulated a complete process of balloon folding and pleating which closely resembled the real scenario. Nonetheless, their study focused on stent expansion and did not investigate the trackability of the balloons. 

In this article, we performed a simulation study using the FEA method to investigate the folding of nylon and Pebax material balloons, as well as the trackability of balloon catheters passing through a 90-degree ideal blood vessel under identical conditions. In addition, a benchtop experiment was conducted to further validate the simulation results.

## 2. Materials and Methods

### 2.1. Bench Tests

The bench test offered clear insights into the performance of the balloon dilatation catheter within tortuous anatomical structures. The test evaluated the insertion forces of folded balloon catheters made of different materials as they navigated grooves—a simulation of the transport conditions within actual curved vessels. The quality of trackability can be evaluated based on the measured insertion force. Therefore, bench tests can provide a reference for balloon design and clinical selection [13].

The tests were conducted using a test tracking fixture (Figure 1), as specified in ASTM F2394-07 [13]. It simulated the bending shapes of coronary arteries on a two-dimensional plane without considering the shapes of lesions. It was composed of two plates, one above and one below, with grooves representing the curved vessels. The grooves featured varying degrees of curvature at different positions. The folded balloon was pushed into the groove through the entrance. A multi-segment displacement load was applied to the balloon: the total push distance was set at 40 mm, the individual push distance at 5 mm, and the push speed was 2 mm/s. After each movement, the subsequent push was executed at regular 2.5 s intervals. A force-measuring device captured the push force exerted by the balloon as it navigated the bend, as depicted in Figure 1.

### 2.2. Materials 

Nylon-12 (polyamide) and Pebax are the two materials most commonly used in balloons [15,25,26]. Two materials were used in this study: Grilamid L25 (nylon 12) from EMS and Pebax^®^ 7033 SA 01 MED resin (a thermoplastic elastomer from Arkema). They are specially designed to meet the stringent requirements of medical applications such as minimally invasive devices, and they both exhibit good biocompatibility [26]. The nylon stress–strain curve was redrawn from a Grilamid L25 technical data sheet. The stress–strain relationship of Pebax is shown in [26]. An elasto-plastic material with an arbitrary stress as a function of strain curve and an arbitrary strain rate dependency can be defined using MAT_89 in LS-DYNA [27]; the model was selected, and the stress–strain curve was input to fit the material properties. The densities, moduli of elasticity, and Poisson’s ratios of nylon and Pebax [15] are shown in Table 1. Clearly, nylon-12 is the stiffer of the two materials. The shaft is composed of high-density polyethylene (HDPE), which was modeled as linear elastic, as per Table 1 [28].

*MAT_89 used the Cowper–Symonds constitutive model [29]. The equation of this model is:(1)σs=σ01+ε˙C1p

Taking the logarithm of both sides of the equation and simplifying it, we obtain:(2)$$\log_{10}\dot{\varepsilon }=\log_{10}C+P\;\log_{10}\left(\frac{\sigma_{s}}{\sigma_{0}}-1\right)$$
where $\sigma_{s}$ is the yield stress of the material, $\sigma_{0}$ is the quasistatic (0.001 s^−1^) yield stress of the material, $\dot{\varepsilon }$ is the strain rate of the material, and *P* and *C* are material constants determined by test [30].

### 2.3. Geometric Models

During the process of manufacturing balloon catheters, cylindrical balloons are first compressed into a folded shape using a balloon-folding machine. Then, they are encapsulated into a smaller diameter using a balloon-crimping machine. Therefore, the balloons are in a folded state before inflation. In order to make the simulation more realistic, the crimping process, which involves the generation of pre-stress in the balloon, should be taken into account.

The balloon model is derived from the Euphora balloon dilation catheter (Medtronic, Minneapolis, MN, USA), featuring a principal diameter of 4 mm and a distal shaft diameter of 0.91 mm. The balloon’s thickness is 0.02 mm [18] (Figure 2). The pleating tool, as shown in Figure 3, has an inner diameter of 2 mm. This tool functions by initially compressing the balloon into three wings and subsequently facilitating each wing’s rotation around the axis, thereby inducing the regular folding of the balloon. 

In order to more accurately simulate the working process of a balloon-crimping machine, the simplistic cylindrical model that was previously widely used was replaced with a novel design composed of 8 plates combined in a specific formation. They formed a chamber that could be enlarged or reduced (as shown in Figure 4). The three-winged balloons were folded and compressed in this chamber, and the encapsulated balloons’ final diameters were then determined.

A geometry model, referred to as the “pipeline”, featuring a groove shape consistent with the ASTM F2394-07 bench test fixture, was designed with a diameter of 1.5 mm [13] (Figure 5). The model features a 90-degree angle between the inlet and outlet, replicating the grooves traversed by the balloon during the force-monitoring phase of the bench test (Figure 1). To prevent mesh distortion and convergence failure due to the pipeline’s edges being cut by balloons, a transitional outward extension was created at the inlet (Figure 5). The outlet was bent 90 degrees relative to the inlet, spanning a total length of 50 mm. The pipeline was modeled as a rigid body and fixed in space.

### 2.4. Finit Element Modeling and Boundary Conditions

The pleating tool moved toward the axis by a distance of 2 mm, resulting in the balloon being compressed into a three-fold configuration (three-wing configuration). Then, it rotated along the axis, causing the three-fold to rotate counterclockwise by 120 degrees. Given that both ends of the balloon were secured to the shaft and a specific negative pressure was exerted on the balloon’s inner surface at that moment, the balloon tightly adhered to and was fixed onto the shaft, while the three outward folds rotated with the pleating tool and attached to the shaft.

At the initial moment, both ends of the balloon were fixed to the shaft. After the pleating tool compressed the balloon, a negative pressure of 6.5 atm was applied to the inner surface of the balloon [23], causing it to contract. The degrees of freedom of the shaft in the x and y directions were fixed, and a continuous displacement load of 20 mm was applied to one end of the shaft. The reason for not employing multi-segment displacement is that when replicating the experimental conditions, the abrupt acceleration and deceleration of the shaft can lead to a significant dynamic effect [31], leading to a loss of the realism of the balloon’s shape.

The contact between the balloon and the pipeline was set to have an ideal friction coefficient of 0.02 [32], while the shaft and the pipeline were set to have no friction. To make the balloon adhere more closely to the shaft, the friction coefficient between the balloon and the shaft was set to 0.2 so that a certain amount of friction force would be generated between them after the negative pressure was applied, preventing excessive distortion of the balloon at both ends during rotation and folding.

A mesh convergence study was conducted to ensure that the calculation results were not significantly affected by variations in the number of mesh elements [33]. In this study, a mesh convergence study was performed for the balloon. Four models with different mesh sizes were created with 54,177, 150,156, 213,539, and 336,542 elements, corresponding to mesh sizes of 0.1 mm, 0.06 mm, 0.05 mm, and 0.04 mm, respectively (Table 2). The models were assigned the material properties of Pebax and subjected to the same simulations of folding, crimping, and insertion into the pipeline. The maximum stress on the balloon surface at the minimum crimped diameter was recorded for each model (Figure 6). The results show that the maximum stress difference between the third and fourth models is less than 5%, indicating convergence and satisfying the criteria for mesh convergence. Therefore, the mesh size of the third model was chosen for subsequent simulations, ensuring computational accuracy without excessive computational time due to redundant mesh elements.

Based on the mesh convergence study, the balloon model was discretized into four-node, fully integrated shell elements with five through-shell thickness integration points. It was discretized using 213,539 mixed shell elements with a thickness of 0.02 mm, and the shaft was discretized using 34,438 mixed solid elements. All other components were set as rigid bodies, and their mesh densities had little effect on the simulation results.

The model discretization was performed using ANSA v21.0 (BETA CAE Systems, Switzerland), and the simulations were performed on 16 CPUs of an AMD EPYC 7532 (GHz) workstation using LS-DYNA Release 13 (LSTC, Livermore, CA, USA).

## 3. Results

### 3.1. Bench Tests

Figure 7 presents the results of the insertion force tests for the nylon and Pebax balloons. The force–displacement curves, characterized by several peaks due to discontinuous loading, demonstrate that the insertion forces of both balloons initially increase and then decrease with greater displacement. The maximum insertion force for nylon is 0.866 N, substantially greater than that of Pebax, which is only 0.156 N.

### 3.2. Numerical Results

#### 3.2.1. Dynamic Folding and Delivery Process of the Balloon

The pleating tool compressed the balloon from its initial form (Figure 8a) into a tri-wing shape (Figure 8c), with the three wings remaining in an inflated state. Upon the application of negative pressure to the balloon’s inner surface, the three wings tightly adhered to each other (Figure 8d). As the pleating tool rotated counterclockwise, it dove the three wings (Figure 8e), which then sequentially attached to the shaft (Figure 8f). Subsequently, the simulated balloon-crimping machine continuously constricted the chamber, yielding a fixed balloon diameter of 1.5 mm. Following this, the balloon was axially inserted into the pipeline to test the insertion force (Figure 9). The balloon’s shape within the pipeline demonstrated that the bent balloon exhibited evenly spaced wrinkles on the concave side Figure 10), causing the adjacent area to bulge beyond the original diameter.

#### 3.2.2. Balloon Stress–Strain Analysis

The stress distribution and effective plastic strain information can help us understand how the balloon behaves under different loading conditions, such as a folding state or bending state. This allows for an evaluation of the balloon’s mechanical integrity and the identification of any areas prone to excessive stress or deformation. After radially crimping the balloons to achieve the same diameters, the stress levels of the nylon balloon were found to be higher than those of the Pebax balloon. The maximum stress was located at the proximal end, with 61.9 MPa for the nylon balloon and 55 MPa for the Pebax balloon, which is consistent with the results of previous studies [24]. The post-folding stress distribution in both balloons was more concentrated at the edges of the three folds and at the balloon ends (Figure 11), where the deformation exceeded that of the remaining areas. Figure 12 more clearly illustrates that higher effective plastic strains consistently manifested at the folds. Within the same fringe range, the Pebax balloon’s middle showed a greater effective plastic strain than the nylon balloon, with the Pebax balloon reaching a maximum effective plastic strain of 0.948, which is higher than the nylon balloon’s 0.879.

#### 3.2.3. Trackability

The insertion force between the balloon and the pipeline was monitored during the process of pushing the balloon into the pipeline (Figure 13). The balloons of both materials showed a gradual increase in insertion force with the increase in pushing distance, and the insertion force reached its maximum value after the balloon was pushed into the pipeline completely at 2.16 ms. Nylon had a maximum insertion force of 1.016 N, while Pebax had a maximum insertion force of 1.021 N.

Similar to the experimental results, the simulation results also exhibited a gradual increase in the force value for the initial 20 mm of the inserting displacement. In this study, the simulation was conducted specifically for the process of the balloon being inserted into the pipeline rather than the process of the balloon being pushed out of the pipeline. Therefore, unlike in the experiment, the contact force did not decrease after reaching its maximum value. In the experiment, the insertion force of the Pebax balloon was significantly lower than that of nylon. However, in the simulation, the insertion force of nylon was only slightly different and higher than Pebax’s at certain moments. The maximum insertion force of the nylon balloon in the simulation was slightly larger than the experimental value of 0.866 N.

#### 3.2.4. Surface Energy Density

The surface energy density provides valuable information about the interaction between the balloon material and the surrounding environment. It can affect the balloon’s ability to navigate through blood vessels, interact with the vessel wall, and perform its intended functions effectively. The surface energy density of the pipeline with balloons fully inserted is shown in Figure 14. Throughout the insertion process, the surface energy density continued to increase and was primarily distributed on the convex surface of the pipeline and the folds of the balloon. The balloon’s surface energy density is significantly higher than that of the pipeline. The Pebax-balloon-loaded pipeline had higher surface energy density than the nylon-balloon-loaded pipeline.

## 4. Discussion

With the same shaft, the nylon balloon exhibited a greater insertion force than the Pebax balloon in the experiments. In the simulations, the coefficient of friction between the balloon and the pipeline was set to the same value for both cases for better variable control [32]. The simplification of the friction coefficient only affects the numerical values of the insertion force, while the comparison of stress and strain remains unaffected by it. The frictional force originated from both the balloon’s tendency to expand outward in its folded state, exerting pressure on the pipeline wall, and from the pressure exerted on the wall as the balloon bent along the pipeline. Although the friction force is also related to velocity [34], the same boundary conditions are imposed on both so that the effect of velocity can also be neglected. The use of the same friction coefficient also results in a less significant difference in the insertion force between the two materials in the simulation compared to the experiment [35]. The difference between the experimental result of 0.866 N for the nylon balloon and the simulated result of 1.016 N can be attributed to the given friction coefficient. The friction coefficient used in the simulation was obtained from previous studies [32] and may not perfectly match the friction coefficient in our actual experimental environment. However, using the same friction coefficient, controlled for the variables, allows for a better assessment of the impact of material stiffness on the insertion force without the need to consider variations in surface characteristics such as roughness.

The experimental results seem to indicate that the stiffer balloon material has more friction during insertion and requires more insertion and retraction forces because it exerts more pressure on the wall than the softer material. However, numerical simulations revealed a negligible difference in the insertion forces between the two materials, contrary to the experimental results. This suggests that the influence of material stiffness on the balloon insertion force is minimal when surface characteristics are disregarded. Therefore, the significant disparity in the insertion force test results between the nylon and Pebax balloons in the experiment is attributed to the differences in their surface characteristics rather than nylon being harder and thus requiring greater force and Pebax being softer and thus requiring less force. This finding is counterintuitive.

Both experiments and simulations demonstrated that as the balloon progresses deeper into the pipeline and subsequently retreats, the frictional force initially increases and then decreases. This suggests that the friction force during the insertion and retraction processes correlates with the length of the balloon within the pipeline. Thus, selecting balloons of different lengths may be necessary to ensure that the friction force during insertion/retraction is not too high for different patients [36,37]. Our future research will explore the use of balloon models of the same material but different lengths to simulate the friction force through grooves, which can provide a reference for appropriate balloon selection.

After balloon folding, the deformation near the connection with the shaft is substantial. The element with the highest stress also occurs in this area, suggesting that this region dictates the overall balloon’s stress concentration level [24]. This implies that a more thoughtful optimization of this area’s structure could potentially reduce the maximum stress experienced after balloon folding. In this study, the proximal and distal folds exhibit slight differences. It is well documented that the elements with the maximum stress are at the distal end of both balloons. This is because the distal diameter of 0.91 mm is larger than the proximal diameter of 0.6 mm, and the transition from the balloon to the connection is steeper at the distal end, leading to the formation of deeper folds (Figure 15). Therefore, a smoother transition at the connections between balloons and shafts, particularly at the neck, could reduce the maximum stress of the balloons after folding.

Wrinkling arises when the balloon undergoes compression on its concave side while passing through the pipeline instead of stretching on the convex side. If the bend only stretches the balloon on the convex side, it only changes the length of the balloon along the axial direction without negatively affecting the balloon’s deliverability. However, squeezing the balloon on the concave side will cause some folds to exceed the original diameter, resulting in an increased cross-sectional profile of the balloon during bending. This condition is unfavorable for navigating through tortuous anatomy [38] and can negatively impact the balloon insertion force, deliverability, and flexibility. To improve balloon delivery, the folded balloon should stretch on the convex side rather than being squeezed on the concave side when bending along the axial direction. This can be achieved by designing a fold structure to be less prone to wrinkling or by fabricating the balloon from a material characterized by superior compression resistance and ease of stretching.

In this study, the assumption of using the same friction coefficient for both materials in contact with the pipeline is a simplified model. In reality, the contact properties between different materials may differ, which is a limitation of this study. In addition, fluid–structure interactions were neglected, as in previous studies [39,40,41] that simulated stent implantation in blood vessels with inflated balloons. These factors need to be considered in future work.

## 5. Conclusions

This study simulated the folding process of two distinct balloon materials and inserted the folded balloons into a simulated pipeline to monitor the insertion forces. Experimental tests were conducted on both types of balloons to measure the actual insertion forces. Through comparing the experimental data and numerical simulation results, it was found that under the assumption of the same friction coefficient in the numerical simulation, the difference in insertion force between the two materials was insignificant. This contrasted with the experimental results in which the harder nylon balloon demonstrated a significantly higher insertion force in comparison to the Pebax balloon. This indicates that the impact of material stiffness on the balloon insertion force is minimal when the surface characteristics of the materials are not considered. Therefore, in practical production and application, for balloons with poor trackability, emphasis should be placed on improving surface smoothness, such as by adding lubricious coatings. Additionally, the stress and effective plastic strain results after folding showed that the harder nylon balloon experienced greater stress after folding, while the softer Pebax balloon experienced a higher level of effective plastic strain. This suggests that nylon balloons are likely to exert greater pressure on the curved vessel wall. This simulation of balloon folding and insertion forces offers a generalized methodology for simulating the trackability of balloons made from different materials. This would be advantageous for selecting suitable balloon materials for various scenarios and may provide valuable insights for the design of balloon structures and folding methods.

## Figures and Tables

**Figure 1 jfb-14-00312-f001:**
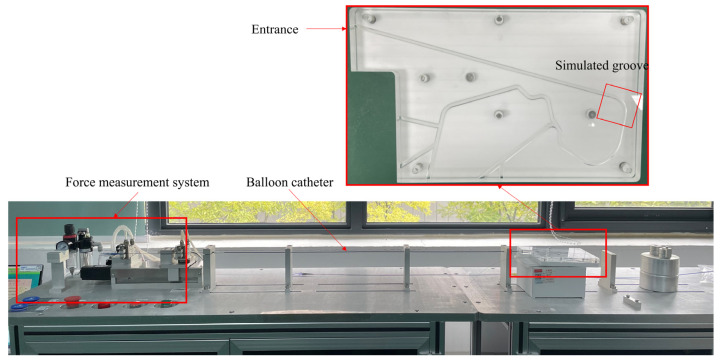
Bench test tracking fixture.

**Figure 2 jfb-14-00312-f002:**
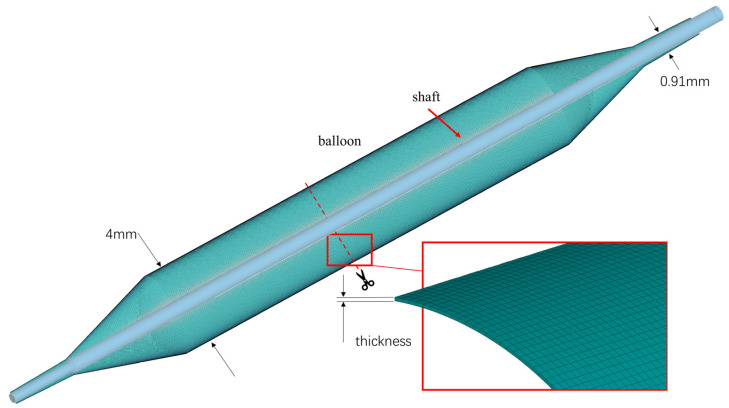
Balloon and shaft.

**Figure 3 jfb-14-00312-f003:**
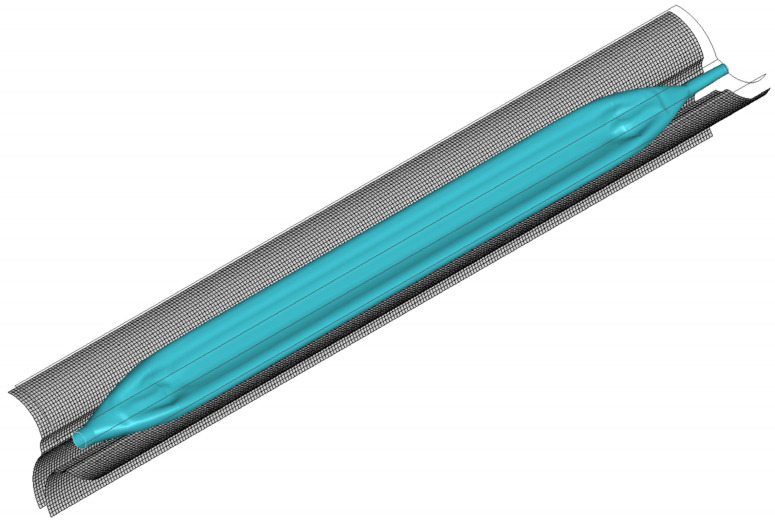
Pleating tools.

**Figure 4 jfb-14-00312-f004:**
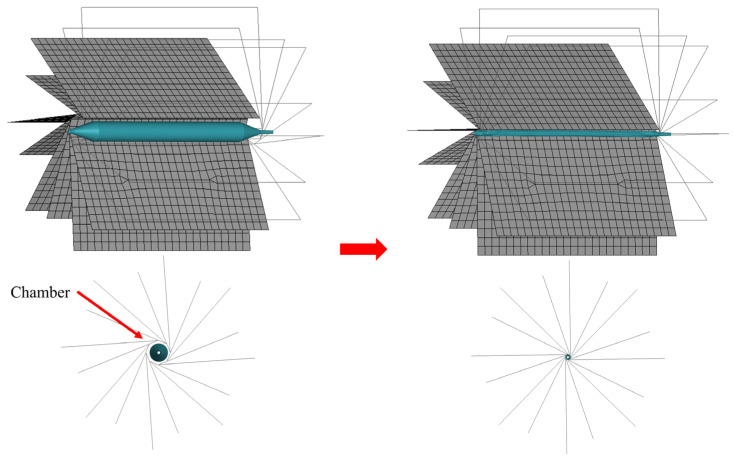
Balloon-crimping simulation.

**Figure 5 jfb-14-00312-f005:**
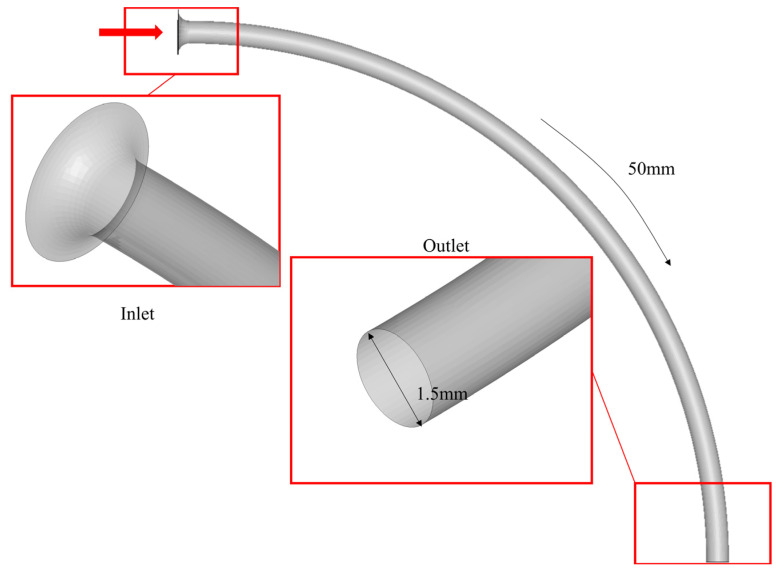
Pipeline. After folding, the balloon will be fed into the inlet.

**Figure 6 jfb-14-00312-f006:**
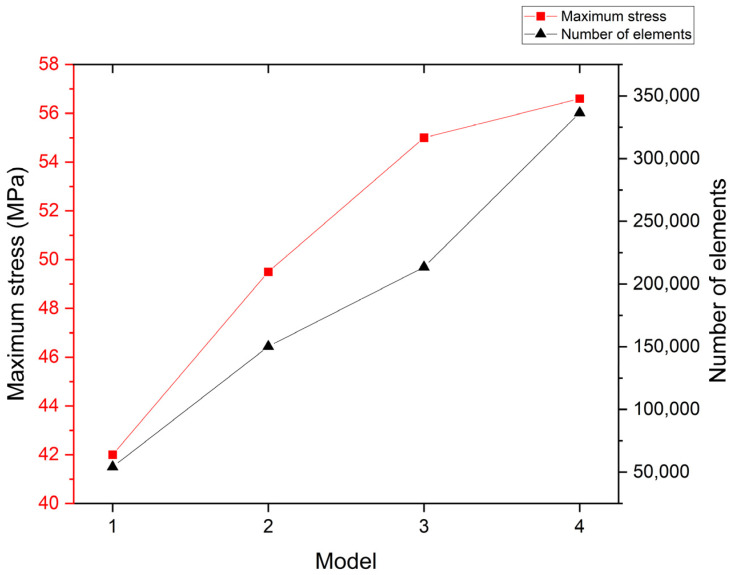
The variation in maximum stress with respect to the number of elements in the mesh convergence study.

**Figure 7 jfb-14-00312-f007:**
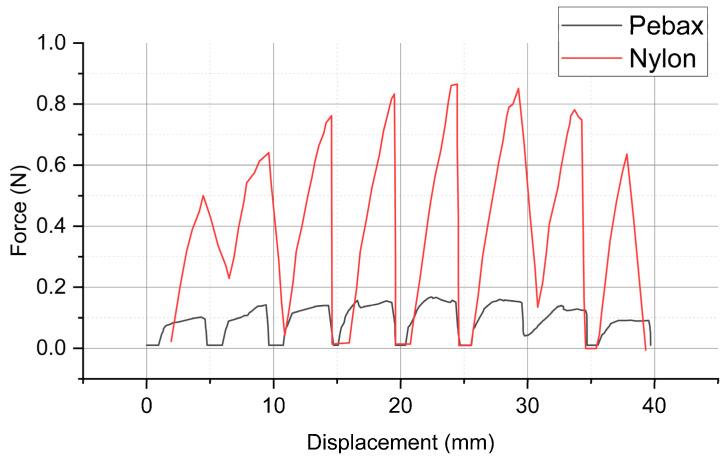
Test results of insertion forces for two types of balloon materials: Grilamid L25 (nylon 12) and Pebax^®^ 7033 SA 01 MED.

**Figure 8 jfb-14-00312-f008:**
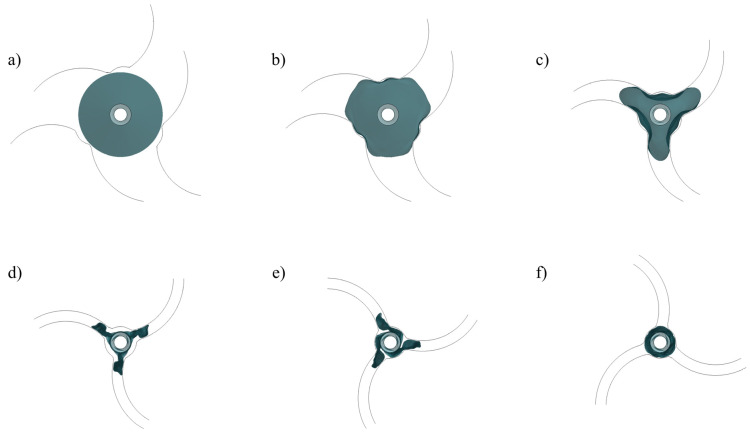
Pleating process. (**a**–**c**): Compression of the balloon by the pleating tool. (**d**–**f**): The three wings are folded onto the balloon.

**Figure 9 jfb-14-00312-f009:**
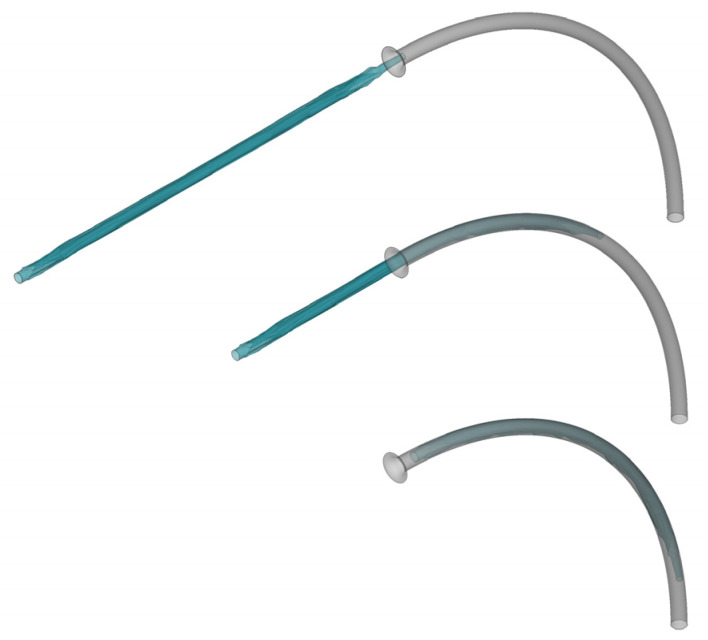
The process of inserting the folded balloon into the pipeline.

**Figure 10 jfb-14-00312-f010:**
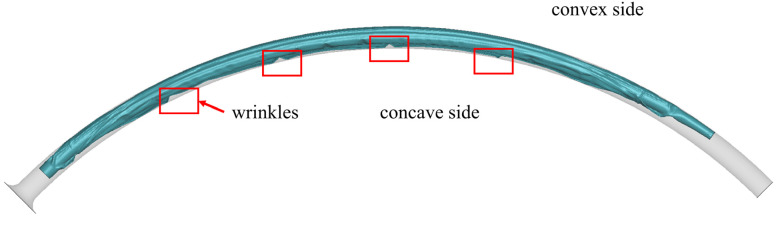
Equally spaced wrinkles in the concave side.

**Figure 11 jfb-14-00312-f011:**
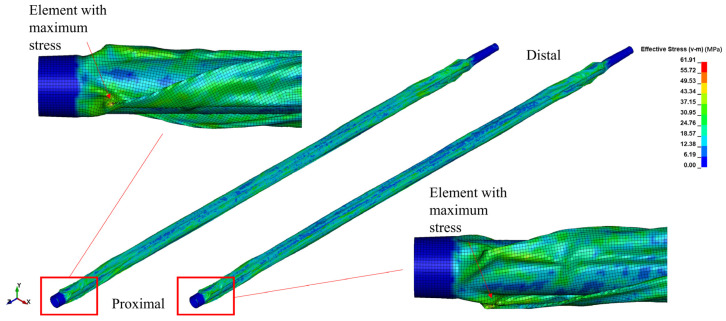
Stress distribution and location of elements with maximum stress after folding for the two types of balloon materials, nylon (**left**) and Pebax (**right**).

**Figure 12 jfb-14-00312-f012:**
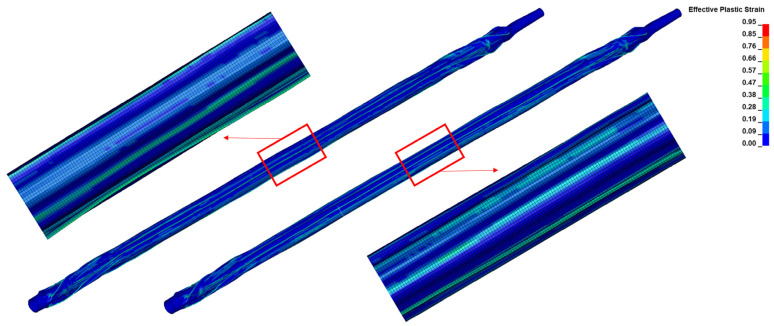
The effective plastic strain fields of the two types of balloon materials: Nylon (**left**) and Pebax (**right**).

**Figure 13 jfb-14-00312-f013:**
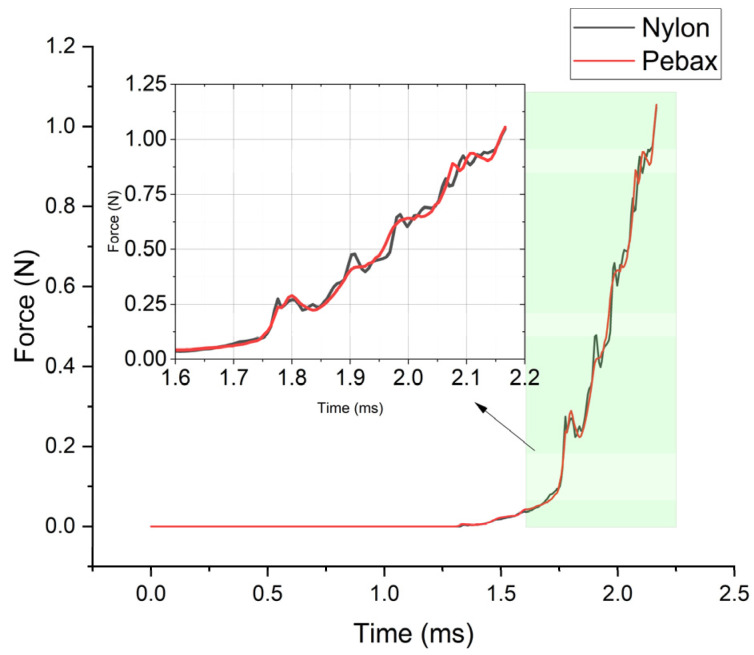
Temporal variation in contact force for two types of balloons.

**Figure 14 jfb-14-00312-f014:**
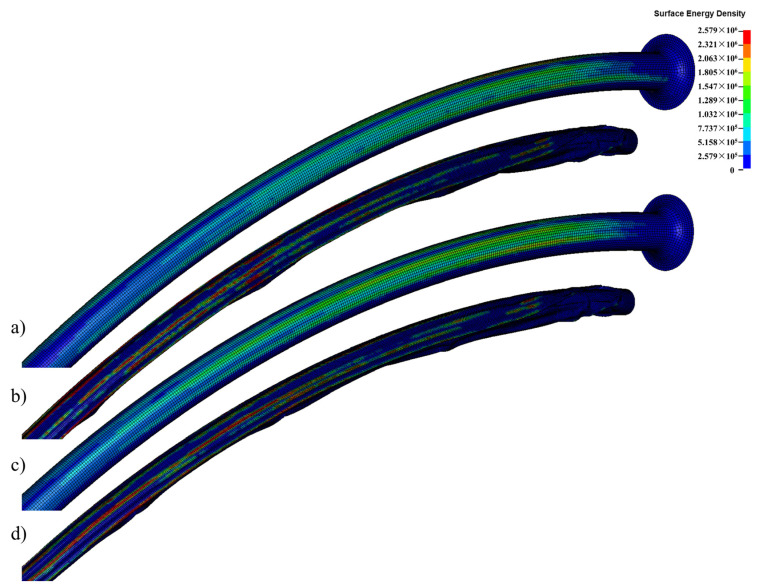
Surface energy densities of two balloons: (**a**) the pipeline loaded with a nylon balloon; (**b**) Nylon balloon. (**c**) The pipeline loaded with a Pebax balloon; (**d**) Pebax balloon.

**Figure 15 jfb-14-00312-f015:**
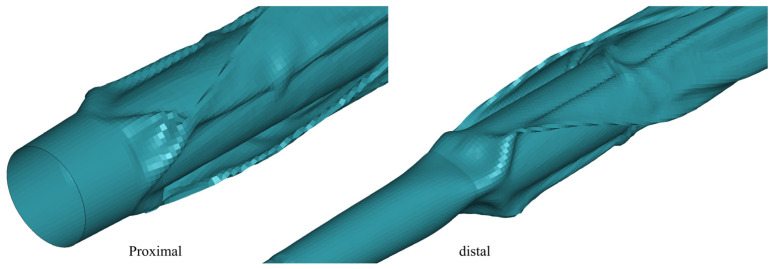
Comparison of balloon folds.

**Table 1 jfb-14-00312-t001:** Material properties.

Material	Density (g/cm^3^)	Young’s Modulus (MPa)	Poisson’s Ratio
Nylon	1.01	1100	0.4
Pebax	1.01	414	0.4
HDPE	0.953	1000	0.46

**Table 2 jfb-14-00312-t002:** Mesh convergence study.

Model	1	2	3	4
Mesh size (mm)	0.1	0.06	0.05	0.04
Number of elements	54,177	150,156	213,539	336,542
Maximum stress (MPa)	42.0	49.5	55.0	56.6

## Data Availability

The data presented in this study are available on request from the corresponding author.

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
