# Peer review of "Simulation and Experimental Investigation of Balloon Folding and Inserting Performance for Angioplasty: A Comparison of Two Materials, Polyamide-12 and Pebax"

_jfb, 2023, doi:10.3390/jfb14060312_

Round 1

Reviewer 1 Report

In this work, the authors have presented a simulation study on the effects of two different materials (PA12 and Pebax) on the flexibility and insertion force of balloon catheters passing through a 90-degree curved pipe under identical conditions. Benchtop experiment per ASTM F2394-82 07 has been conducted to further validate the simulation results. Following are some comments for consideration:   

1)    Line 34/35: Elaborate on different roles of balloons

2)    Line 68/69: Although simulations can help minimize experimental efforts, it cannot help eliminate the need for experiments. Regulatory bodies like the FDA will always require bench testing data to support safety and efficacy of the device. Please consider updating this accordingly.  

3)    Lines 100 and 102: Correct ‘reference not found’ errors   

4)    Use clearer stress/strain scales for all simulation figures

5)    In the discussions section, explain the reasons for differences seen in the force values obtained in simulations vs those obtained in bench tests

6)    Several English language improvements should be made. A few examples are provided below.  

a.     Typo in Title -- Pebax , not Pebex

b.     Line 33: use a number describe population size

c.     Line 36: Correct Figure 4 caption

See above. 

Reviewer 2 Report

Simulation and experimental investigation of balloon folding and inserting performance for angioplasty: a comparison of two materials – PA12 and Pebax

In this study, the authors used computational and experimental approaches to investigate the differences between two common materials used for balloon catheters, Nylon-12 and Pebax. While the methods used are appropriate, the manuscript can be improved to better convey the importance of the study and clarify various aspects of what was done and why. The following are some specific comments that should be addressed before consideration for publication.

·       Title: Typo “Pebex”; consider changing PA12 to Polyamide-12 or Nylon, since this is how it is referred to in the manuscript.

·       Abstract: objective of the study isn’t stated; need to be proof-read, some sentences are unclear and doesn’t provide specific information i.e. it is vague. Methods should reflect the actual model created, and experimental and simulation parameters. What is meant by relevant research? What exactly does this information add/contribute? How is the simulation more precise and detailed compared to benchtop experiments?

·       Methods indicate the authors were knowledgeable and considered all relevant aspects during study conceptualisation/planning, however this needs to be translated into a clear, impactful manuscript. Check entire manuscript for typos, sentence structure, ensure message is clear and concise. Currently, importance is lost amidst wordy and unnecessary explanations. For metrics investigated – why were these selected, importance. For findings – how does this contribute to the angiography procedures?

Check entire manuscript for typos, sentence structure, ensure message is clear and concise. 

Reviewer 3 Report

In this article, the authors utilize FEA method to presents a simulation study on the effects of two different materials: Nylon and Pebax on the flexibility and insertion force of balloon catheters passing through a 90-degree curved pipe under identical conditions. The manuscript given several comments that needs to addressed.

1.    The present study only compared two different materials consist of PA12 and Pebex? Other materials would be added to enrich the data and improved the present article.

2.    What is the Novel of present computational simulation? The reviewer does not seem any something really new in performed study. Lack of novel needs it should be rejected.

3.    Previouse balloon folding should be explained to shows present state of the art.

4.    See line 100 and 102, there is an error? Please check it.

5.    Line 159-160, the authors informed about convergence study. However, the brief explanation of this study does not given. It is performed by running identical models with different elements, including coarse elements (small number of elements) and fine elements (large number of elements). It aims to select computational models with an optimal number of elements, which does not use too many elements in order to not burden the computational load, whilst still being able to provide accurate results. The authors should give this information. Also, the authors invited to read and incorporated relevant reference as follows, doi: 10.3390/biomedicines11030951, 10.3390/su142013413, and 10.3390/ma16093298

6.    Line 76, why bench test used? There is other test that possible to conduct other that it. Any explanation? Provide this urgency in the revised form.

7.    In Figure 9 and 10, where is maximum place of the contour? Shows it.

8.    Related to convergence study, please proved the graph of convergence study consisting of results obtained and element used for obtained the results.

-

Round 2

Reviewer 2 Report

Objective still wasn't included in the abstract.

While the authors provided further explanations, the scientific writing can be further improved. Note: the use of "wildly" - not sure if this is intentional or a typo.

Must be improved before publication.

Reviewer 3 Report

Crucial comments needs to clarify along with recommendation given in this stage.

1.      Line 109 for bench test tracking fixture. Please extend the explanation for this section.

2.      Line 117,” MAT_89/*MAT_PLASTICITY_POLYMER: it should not the correct way to mentioned it int the article. Not a code, should be explicit name.

3.      Line 118-120 has explained materials properties, but it also given in table 1. The reviewer think it is such a redundant. Please chose one.

4.      Line 182, it stated “were set to have no friction”, how? It should be having a friction. Please for the explanation.

5.      Line 187-199, related the additional explanation given to the authors explaining mesh convergence study. Please incorporated other two literature in previous review report given since it is related.

6.      The biocompatible aspect of biomaterials used needs to stated to give clear understanding to the reader. It is crucial to ensure there is negative response from tissue.

7.      The authors should explain the advantages of computational simulation compared to experimental investigation in functional biomaterials. There would be from faster results and lower cost.

-

Round 3

Reviewer 3 Report

Thanks to the authors for revised version. The reviewer given minor comments in this form.

1.      lines 80-83, related to advantages of computational simulation compared to experimental investigation, give the supporting reference as follows, doi: 10.1016/j.heliyon.2022.e12050, 10.1038/s41598-023-30725-6, and 10.3390/met12081241

2.      Lines 120-122, related the urgency of biocompatible aspect of biomaterials. Incorporated relevant previous study as follows, doi: 10.3390/ma14247554, 10.3390/jfb13020064, and 10.3390/jfb12020038

-